# Global sinusoidal seasonality in precipitation isotopes

Scott T. Allen[1], Scott Jasechko[2], Wouter R. Berghuijs[1], Jeffrey M. Welker[3,4], Gregory R. Goldsmith[5], James W. Kirchner[1,6,7]

[1]Department of Environmental Systems Science, ETH Zurich, Zurich, 8092, Switzerland.
[2]Bren School of Environmental Science and Management, University of California at Santa Barbara, Santa Barbara, CA, 93117, USA
[3]Ecology and Genetics Research Unit, University of Oulu, Finland and UArctic
[4]Biological Sciences Department, University of Alaska, Anchorage
[5]Schmid College of Science and Technology, Chapman University, Orange CA, 92866, USA
[6]Swiss Federal Research Institute WSL, Birmensdorf, 8903, Switzerland
[7]Department of Earth and Planetary Science, University of California, Berkeley, California, 94709, USA

*Correspondence to*: Scott T. Allen (scott.t.allen@utah.edu)

**Abstract**: Quantifying seasonal variations in precipitation $\delta^2$H and $\delta^{18}$O is important for many stable isotope applications, including inferring plant water sources and streamflow ages. Our objective is to develop a data product that concisely quantifies the seasonality of stable isotope ratios in precipitation. We fit sine curves defined by amplitude, phase and offset parameters to quantify annual precipitation isotope cycles at 653 meteorological stations on all seven continents. At most of these stations, including in tropical and subtropical regions, sine curves can represent the seasonal cycles in precipitation isotopes. Additionally, the amplitude, phase, and offset parameters of these sine curves correlate with site climatic and geographic characteristics. Multiple linear regression models based on these site characteristics capture most of the global variation in precipitation isotope amplitudes and offsets; while phase values were not well predicted by regression models globally, they were captured by zonal (0°-30° and 30°-90°) regressions, which were then used to produce global maps. These global maps of sinusoidal seasonality in precipitation isotopes based on regression models were adjusted for the residual spatial variations that were not captured by the regression models. The resulting mean prediction errors were 0.49 ‰ for $\delta^{18}$O amplitude, 0.73 ‰ for $\delta^{18}$O offset (and 4.0 ‰ and 7.4 ‰ for $\delta^2$H amplitude and offset), 8 days for phase values in latitudes outside of 30°, and 20 days for phase values in latitudes inside of 30°. We make the gridded global maps of precipitation $\delta^2$H and $\delta^{18}$O seasonality publicly available. We also make tabulated site data and fitted sine curve parameters available to support the development of regionally calibrated models, which will often be more accurate than our global model for regionally specific studies.

## 1 Introduction

Characterizing the stable oxygen ($^{18}$O/$^{16}$O) and hydrogen ($^2$H/$^1$H) isotope compositions of precipitation can provide insights into the temporal and spatial origins of water, and of geological and biological materials that incorporate O and H from water.

However, the isotopic composition of precipitation is difficult and costly to measure across large spatial scales or at high temporal frequencies, and thus precipitation isotope measurements are often unavailable for the times and locations at which they are needed. Consequently, compiled precipitation isotope data (e.g., Global Network for Isotopes in Precipitation; International Atomic Energy Agency) and interpolations of mean and monthly precipitation isotope data (e.g., Bowen et al., 2005; Bowen & Wilkinson, 2002) are used across many fields of science (West et al., 2010).

Although these network datasets and interpolated maps contain spatial and temporal information, it is often convenient to simplify and average across one of those dimensions. When identifying the spatial origin of water in a sample, investigators may use spatial patterns in mean isotope ratios (despite those patterns varying temporally and those samples not integrating water signatures throughout years). Additionally, when identifying the temporal origin of water in a sample, investigators often use time-series of isotope data from the nearest measurement location (and thus do not account for spatial differences). Alternatively, concise representations of large-scale spatiotemporal precipitation isotope patterns could be widely useful and mitigate the need to average precipitation isotope data across space or time. Various tools and interpolation schemes exist for predicting precipitation isotope ratios at a given location (e.g., Online Isotopes in Precipitation Calculator following Bowen and Revenaugh, 2003), or for mapping spatial patterns in mean or monthly values over specified intervals (e.g., Isomap.org following Bowen et al., 2014). However, previous methods have not explicitly supported predictions of seasonal isotope cycles by first using metrics that capture isotopic temporal dynamics and then interpolated those metrics.

Isotope ratios in precipitation often follow distinct seasonal cycles that can be approximated by sine curves (Bowen, 2008; Dutton et al., 2005; Feng et al., 2009; Halder et al., 2015; Vachon et al., 2007; Wilkinson and Ivany, 2002), and the parameters describing those sine curves are often predictable in space (Allen et al., 2018; Jasechko et al., 2016). Sine curves concisely represent temporal dynamics because they express continuous, cyclic time series as functions of only three parameters (*amplitude*, *phase*, and *offset*). To predict isotope seasonality across the globe, values of these three sine parameters, fitted to monthly precipitation isotope data at monitoring stations, can be described as functions of station climate and geography. Such mapped sinusoidal cycles were shown to be effective in predicting monthly precipitation isotope values across Switzerland (Allen et al., 2018). Beyond being useful for predicting isotope values in specific seasons, sine curves generally aid in characterizing the propagation of cyclic signals. For example, as precipitation travels through hillslopes and into streams, seasonal isotope amplitudes are dampened, reflecting transport processes that can be quantified as a stream-precipitation amplitude ratio (Kirchner, 2016a, 2016b); this *young water fraction*, which requires sine curve fitting of precipitation isotopes, has been used in many recent studies (Clow et al., 2018; von Freyberg et al., 2018; Jacobs et al., 2018; Jasechko et al., 2016, 2017; Lutz et al., 2018; Song et al., 2017). Thus, there are immediate applications for mapped sine curves that characterize precipitation isotope cycles across the globe. More generally, spatial data describing how precipitation isotope compositions

vary seasonally could facilitate interpretations of environmental $^{18}O/^{16}O$ and $^{2}H/^{1}H$ data and support predictions of precipitation isotope compositions in time and space.

Here we present global maps of precipitation isotope cycles that capture patterns in precipitation isotope seasonality. We first describe the strength of seasonal isotope cycles, and quantify how well sine curves explain monthly precipitation measurements at each of 653 precipitation isotope monitoring stations. We then explore how well the parameters describing those sine curves can be predicted across the globe, as a function of site characteristics. Lastly, we produce global maps and data that support stable isotope applications, and make these maps and data publicly available. We conduct these analyses to support a growing need for quantifications of seasonal cycles in precipitation isotopes, not to challenge the methods previously used in other precipitation isotope models.

## 2. Methods

### 2.1. Data

We used a global dataset of monthly precipitation oxygen and hydrogen isotope measurements from 650 and 610 precipitation monitoring stations, respectively. These previously compiled (Jasechko et al., 2016) data were collected from the Canadian Network for Isotopes in Precipitation (Birks and Edwards, 2009; Birks and Gibson, 2013), the US Network for Isotopes in Precipitation (Delavau et al., 2015; Welker, 2000, 2012), and the Global Network for Isotopes in Precipitation (Aggarwal et al., 2011; Halder et al., 2015). Following Jasechko et al. (2016), we characterize seasonal cycles only at monitoring stations that report precipitation isotope compositions for at least eight unique months. Monthly precipitation amounts (or snow-water equivalencies) are also available from 623 of the 650 stations that measured oxygen isotope ratios, and from 603 of the 610 stations that measured hydrogen isotope ratios. All hydrogen and oxygen isotope ratios of precipitation are denoted as $\delta^2H$ and $\delta^{18}O$, defined by

$$\delta^2 H = \frac{(^2H/^1H)_{sample} - (^2H/^1H)_{V-SMOW}}{(^2H/^1H)_{V-SMOW}} \times 1000\ \permil\ , \quad (1)$$

and

$$\delta^{18} O = \frac{(^{18}O/^{16}O)_{sample} - (^{18}O/^{16}O)_{V-SMOW}}{(^{18}O/^{16}O)_{V-SMOW}} \times 1000\ \permil\ , \quad\quad (2)$$

where *V-SMOW* refers to the Vienna Standard Mean Ocean Water standard.

We compiled gridded climatological and geographical data for global modelling and for inferring site characteristics of the precipitation monitoring stations (Figure 1). We downloaded climate maps of monthly precipitation sums and monthly means of daily low, high, and mean temperature, all at 5-arc-minute (i.e., 0.083°) resolution (WorldClim; Fick & Hijmans, 2017). Station climate data were inferred from these gridded products for all but three stations that were on small islands or stationary

weather vessels, for which local meteorological data were acquired. The range of mean monthly temperatures was computed at each pixel (and each monitoring station) as the difference between the highest and lowest monthly mean values, using the WorldClim data. Annual mean daily temperature range was calculated as the mean differences between daily minimum and maximum temperatures. The WorldClim data were also used to calculate time of peak precipitation and temperature, and

seasonal amplitude of precipitation and temperature, metrics which can together capture global patterns in hydroclimate (Berghuijs and Woods, 2016). We also used a 30-second gridded elevation map (GTOPO30; US Geological Survey, 1996) that was aggregated to 5-minutes for consistency with the other grids. Monitoring station elevation data were not inferred from the grids, but instead downloaded directly from the isotope network databases. Distance from oceans and seas was calculated in ArcGIS 10.4.1 (ESRI, Redlands, USA) using published coastline data (Wessel and Smith, 1996) for the centre of each 5-

minute pixel and for each monitoring station.

## 2.2. Sine-fitting methods

We fitted sine curves (described by the parameters *amplitude*, *phase*, and *offset*) to each monitoring station's monthly measured $\delta^{18}$O and $\delta^{2}$H time series using a nonlinear fitting routine ("fitnlm" in MATLAB R2016B, Mathworks, Natick, Massachusetts, USA). The sine curve is defined with a fixed period of one year,

15                  Precipitation $\delta^{18}$O or $\delta^{2}$H $(t) = amplitude \times \sin(2\pi t - phase) + offset$ ,    (3)

where $t$ is the fractional year. All fitted amplitudes and phases were adjusted so that fitted amplitude values are positive, and phase values are between $\pi$ and $-\pi$. *Phase* was calculated in radians, but we report all values in days from the summer solstice. Allen et al. (2018) previously confirmed that this non-linear fitting routine yields parameter values and component standard errors that are equivalent to those obtained by fitting sine curves as an additive model of sine and cosine functions with their

uncertainties calculated by Gaussian error propagation. It should be noted that standard errors depend on the length of records, and while some stations have datasets that are as long as 57 years, shorter durations are more common (Figure 2a). We fitted the sine curves by two alternative approaches: a) using iteratively reweighted least squares with a bisquare weighting function (*robust-fitted*), and b) using standard least squares with the influence of each monthly isotope measurement weighted by the amount of precipitation during that month (*amount-weighted*). These metrics have different limitations. The amount-weighted

cycles are less influenced by erratic values that can occur in low-precipitation months, but also do not capture the variations during drier seasons as effectively. For example, if there was an anomalously dry month in a short data record and that dry month also had an atypical isotope value (e.g., because it was composed of a single small event), that value could result in a robust-fit exaggerating the true seasonal isotope cycle. If estimates based on that sinusoid were later weighted with typical precipitation amounts, this could introduce errors. Weighted-fits could introduce errors if drier season precipitation is important

to the study system, but the dry season precipitation has minimal influence on the fits and thus those values are misrepresented. Weighted fits might also mischaracterize the seasonal dynamics of a typical year in regions that are impacted by extreme precipitation in some years (e.g. hurricanes or monsoons) if that extreme precipitation has distinct isotope values and yields

volumes that are substantial fractions of annual precipitation (e.g., Price et al., 2008). We focus on the robust-fitted parameters describing the seasonal cycles, but for comparison, the amount-weighted fits are also reported in Supporting Information 2. We recommend that future users of these data carefully consider the different limitations when selecting between these two approaches.

**2.3 Precipitation sinusoidal prediction methods**

To characterize spatial variations in precipitation isotope seasonality, we establish relationships between the fitted sine parameters (*amplitude*, *phase*, and *offset*) and site characteristics of the precipitation isotope monitoring stations using multiple linear regression. To characterize the monitoring stations, we used elevation, absolute latitude, distance from the nearest ocean, mean annual temperature, range of mean monthly temperatures, seasonal amplitude of precipitation amount, and mean annual

precipitation amount (Figure 1). We chose these metrics as spatial predictors because global datasets of these metrics are publicly available and they capture aspects of climate and circulation patterns that are known to affect precipitation isotopic composition (Aggarwal et al., 2016; Birks and Edwards, 2009; Rozanski et al., 1993). To determine which predictors should be included in regression models, we used a stepwise model selection approach in which different combinations of predictors were used to maximize $R^2$ values while requiring that all coefficient p-values are statistically significant ($p < 0.05$). This step

limits model overfitting by excluding redundant or non-significant predictors. We found that using these criteria more aggressively removed variables compared to the more standard Akaike Information Criterion (AIC). To assess collinearity among these variables, we calculated the variance inflation factors (VIF) associated with a hypothetical model that includes all six variables; we found those factors to range from 1.4 to 7.8, and while no fitted models actually included all six terms, the variance inflation factors among the six predictors are still all below the often-used threshold of 10 (Marquaridt, 1970).

After identifying the appropriate model terms, models were fitted using the "fitlm" function with robust fitting options that reduce the influence of outliers (MATLAB R2016B). In preliminary analyses, we also tested other metrics – precipitation phase, temperature phase, and mean daily temperature range – but determined that they were not consistently important (i.e., when included in the initial model selection, they were mostly excluded). Thus we excluded these other metrics from subsequent analyses to avoid overcomplicating the models; however, they often showed interesting relationships with the sine

parameters, so they are provided in Figure S1.

For models of phase, we only used data from monitoring stations where there is a distinct seasonal cycle, because phase terms are meaningless and fitted values are unstable where there are no sinusoidal seasonal cycles; these phase values will also be excluded from the supporting information data files to avoid confusion. We characterize distinct seasonal cycles as ones where

the phase is well constrained, with standard errors of the fitted phase terms lower than 15 days (and thus 95 % confidence intervals of approximately ± 1 month). Roughly 74 % of the sites (*n* = 479) met this criterion. We also tested other criteria for filtering out stations with meaningless phase terms, such as $R^2 > 0.3$ (*n* = 425) or $R^2 > 0.5$ (*n* = 232), and those yielded similar

regression models for phase. We modelled phase in mid and high latitudes (30° to 90°; $n = 349$ after removing data without distinct seasonal cycles) separately from phase in tropical and subtropical latitudes (0° to 30°; $n = 130$ after removing data without distinct seasonal cycles). We took this approach because initial inspections of these data and past examinations of similar data (Bowen and Revenaugh, 2003; Feng et al., 2009; Halder et al., 2015) suggested that phase is relatively consistent

within each of these zones, with sharp transitions at approximately 30° N and S (roughly corresponding with Hadley Cell boundaries; Birner et al., 2014).

These fitted spatial regression equations for *amplitude*, *phase*, and *offset* were used to map global precipitation isotope seasonality using the gridded site-characteristic data. We did not extend these maps to extrapolate Antarctic isotope seasonality

because there are few monitoring stations there. We also mapped the residuals, estimated by subtracting the regression model estimates of *amplitude*, *phase*, and *offset* from the same variables determined from the fitted sine curves at the precipitation monitoring stations. We interpolated those residuals using inverse-distance weighting of the residual values from the three stations that are most proximal to each grid-cell centre. For phase, we used nearest neighbour interpolation, rather than inverse-distance weighting, because averages across unlike phases are poorly representative. We then applied a Gaussian filter to

smooth each of the residual adjustment layers, with the standard deviation equal to 3°, because we assume there are measurement uncertainties and thus the layer should not be fitted exactly to the points; we smoothed the phase residuals separately in absolute latitudes > 30° versus absolute latitudes < 30°. For final predictive maps, we added the smoothed residual maps to the regression-based maps; wherever negative amplitudes were resulted, those values were forced to zero. Errors were evaluated by running this routine again, but while randomly excluding 65 sites (10%) for subsequent use as independent

quality-control checks. Sine parameters for those 65 stations were predicted using models calibrated with the other ~585 sites; this Monte Carlo procedure was iterated 15 times for both $\delta^{18}$O and $\delta^{2}$H.

We provide these predictive maps of the gridded amplitude, phase and offset values of $\delta^{18}$O and $\delta^{2}$H. We also provide gridded amplitude, phase, and offset values for precipitation amount, which can be used to scale precipitation isotopic inputs, in

applications where amount is important. These maps are provided (Supporting Information 3).

To explore sub-global variations in performance of the spatial multiple regression models, we also performed regional regression analyses in which we fitted multiple regressions to data from subsections of the globe. Regressions of *amplitude*, *phase*, and *offset* were calculated for 40° × 40° windows using the same site characteristics that were used in the global models:

absolute latitude, elevation above sea level, distance from coastline, range of mean monthly temperatures, mean annual temperature, and annual precipitation amount. These regional regressions were calculated at all vertices of a 10° grid (marking the centre of each 40° window). We used the same combination of stepwise regression model selection and robust regression fitting as in the global analysis. Only windows that contained more than 25 precipitation isotope monitoring stations were

analysed. We report gridded $R^2$ and root mean square error (RMSE) values to indicate where these relationships are strongest. We also provide fitted sine parameters and site characteristics in the supporting information to facilitate users' development of other regression models for regionally specific applications (Supporting Information 2).

## 3 Results

### 3.1 Seasonal cycles in precipitation isotopes

Globally, 94 % of the precipitation $\delta^{18}O$ monitoring stations (n = 650) have statistically significant seasonal isotope cycles ($p < 0.05$; t-test of the $\delta^{18}O$), although those cycles do not always explain the majority of the variance in monthly isotope values (i.e., only 36 % of the stations had $R^2$ greater than 0.5; Figure 2). Amplitudes range from 0 to 11 ‰ $\delta^{18}O$ (Figure 3), with a median value of 2.3 ‰ $\delta^{18}O$; here, amplitude quantifies the strength of seasonal cycles as deviations from average annual values, so an amplitude of 2.3 ‰ $\delta^{18}O$ corresponds to a range of 4.6 ‰ between typical values in the "higher $\delta^{18}O$ season" and the "lower $\delta^{18}O$ season". Seasonal isotope variations are larger in colder, higher-latitude, higher-elevation, or more continental regions (Figure 3), although no individual site characteristic explains the majority of variation in amplitude (Figure 3; Table 1). The few coastal stations that have strong seasonal cycles are almost exclusively located in high absolute-latitude regions (Figure 4a). Many of the monitoring sites within tropical latitudes also have substantial seasonal cycles; for example, 27 % of sites in the tropics show amplitudes greater than 3 ‰ $\delta^{18}O$, and they are not all high-elevation sites (Figure 3b).

Although most stations show a seasonal precipitation $\delta^{18}O$ cycle, the ability of sine curves to capture monthly $\delta^{18}O$ values varies (Figure 2). The median percent of variance explained by sine curves is 42 %; the median RMSE of individual monthly deviations from fitted sine curves is 2.2 ‰ $\delta^{18}O$. Stronger fits occur where a) there is a strong seasonal cycle, b) the seasonal cycle is the dominant pattern of variation, and c) sine curves are the appropriate shape to characterize precipitation isotope variations. Accordingly, the spatial pattern in $R^2$ (Figure 2c) is broadly similar to the pattern in amplitude ($r = 0.74$). However, RMSE also increases with amplitude ($r = 0.58$), demonstrating that greater seasonal variability is also generally associated with greater month-to-month deviations from the seasonal sinusoidal cycle.

The phase term is well constrained (i.e., SE of phase < 15 days) at most but not all sites ($n = 479$), and its geographic distribution is surprisingly binary (Figure 4b). From 30° S to 30° N (i.e., roughly corresponding with the Hadley cells), peak isotope values occurred $104 \pm 43$ days before the summer solstice (mean $\pm$ SD). By contrast, in the mid- and high-latitude regions, peak isotope values occurred $18.6 \pm 24$ days after the summer solstice. A few exceptions are found in absolute latitudes near 30°, which may be attributable to the effects of the Asian monsoon cycle (Cai et al., 2018) or the migration of Hadley cell boundaries, which do not consistently occur at 30° (Chen et al., 2014). Peak precipitation isotope values occur within a month of peak temperature at 89 % of the monitoring stations that are in absolute latitudes above 30° and have well-constrained

seasonal isotopic phases (Figure S2); however, that pattern was not ubiquitous. On average, phase of $\delta^2H$ significantly lags $\delta^{18}O$ in absolute latitudes over 30° ($p < 0.01$), albeit with a median difference of only 2 days (and median absolute difference of 4 days); these observations suggest that precipitation LC-excess, defined as $\delta^2H–a×\delta^{18}O−b$ (where $a$ is the slope and $b$ is the intercept of the LMWL; Landwehr and Coplen, 2006), may frequently have a seasonal cycle, as previously described in

Switzerland (Allen et al., 2018) and suggested in global deuterium-excess variations (Pfahl and Sodemann, 2014).

Offset values, describing the central tendency of the seasonal cycle, span a range of 33 ‰ in $\delta^{18}O$. These values are highest (least negative) in tropical and subtropical regions, and lowest in polar regions (Figure 4c). Most prominent is the strong temperature trend (0.47 ‰ $\delta^{18}O$ per °C, $R^2 = 0.77$; Figure 3; Table 1), consistent with patterns that have been previously

described (Dansgaard, 1964; Rozanski et al., 1993). It should be noted that offsets and amplitudes are associated differently with continentality (Figure 4 a,c); while many of the regions with highly negative offsets also have large amplitudes, this is untrue of coastal regions in mid and high latitudes where highly negative offsets and small amplitudes co-occur. For example, in Reykjavik, Iceland, the $\delta^{18}O$ offset is -8.0 ‰ and the amplitude is 0.9 ‰; a similar offset is found in continental Iowa, USA (-8.2 ‰), but the amplitude is 4.5 times larger (4.0 ‰).

### 3.2 Spatial patterns in parameters describing precipitation isotopic cycles

The spatial patterns in amplitude, phase, and offset can be described as functions of site characteristics. Of the predictors examined, all have significant correlations (at $p < 0.05$) with amplitude, phase, and offset (Table 1; see also Figure 3). Spearman rank correlations, which are less influenced by extreme values, are also statistically significant for all but one of these

relationships (Table 1). However, no variables explain the majority of variation in amplitude, and only temperature explained the majority of variation in offsets (Table 1).

We developed multiple linear regression models of site characteristics and sine parameters, and used them to generate maps of $\delta^{18}O$ sinusoidal cycles (Figure 4). The multiple regression models explain 64 % of the variation in amplitude (RMSE = 1.1

‰) and 83 % of the variation in offset (RMSE = 2.0 ‰). The multiple regression models for phase have low $R^2$ values (0.19 and 0.21, respectively for absolute latitudes above and below 30°) because there is little variation in phase within each latitude band; thus, phase RMSE values are small (12 and 28 days; Table 2). The coefficients of the multiple regression equations describing mapped precipitation $\delta^{18}O$ sinusoidal cycles are presented in Table 2 and analogous coefficient tables describing global regression models of $\delta^2H$, amount-weighted $\delta^{18}O$, and amount-weighted $\delta^2H$ cycles are presented in Table S1.


Residuals from the interpolated sine parameter layers are often show clusters of similar values (Figure 5), implying that sources of geographic variation are not fully captured by the predictors that we have used. Consequently, regionally calibrated models

(calculated over moving 40° × 40° windows) often yield better fits (Figure 6). Even in regions where multiple regression models do not effectively explain the variations in precipitation isotope sine parameters (e.g., Central America, South-central Asia), they will necessarily be fitted to the mean regional values, so the regional multiple regression model errors (RMSEs) will usually be smaller than those of the global regression model.

To produce final predictive maps, we adjusted for the geospatially clustered residuals by adding the smoothed residual maps (Figure 5) to the regression-based maps (Figure 4). These predictive sinusoidal maps of $\delta^{18}O$ seasonality (Figure 7) and $\delta^2H$ seasonality (Figure S3) are made available in the supplementary materials. They capture 88 %, 97 %, and 96 % of the global variations in amplitude, phase, and offset, respectively. To calculate the prediction errors, we ran this routine again, but

randomly excluded 10% of the sites from the calibration so that the sine parameters at those sites were predicted independently; the median amplitude and offset errors were 0.49 ‰ and 0.73 ‰ $\delta^{18}O$ (and 4.0 ‰ and 7.4 ‰ $\delta^2H$), and median phase errors were 8 and 20 days (respectively for absolute latitudes above and below 30°).

**4 Discussion**

The occurrence of seasonal cycles in precipitation isotopes enables tracking how precipitation cycles propagate through

landscapes and ecosystems. Previous research has found that precipitation isotopes vary seasonally, and that these seasonal patterns vary geographically (Halder et al., 2015; Rozanski et al., 1993). This work quantifies those seasonal patterns and their geographical variation, yielding global maps of sinusoidal precipitation isotope cycles (i.e., global sinusoidal 'isoscapes') that can be used to predict seasonal precipitation isotope cycles in sites or regions where they have not been measured.

Site characteristics explain most of the global precipitation isotope cyclicity, albeit with uncertainty in the regression model, the sine fits, and the raw data. Amplitude variations are mostly predictable by multiple regression (Table 2), but there were regional clusters of substantive (±1-2 ‰ $\delta^{18}O$) amplitude residuals. For example, the regression model (Figure 4) tended to systematically underestimate amplitudes in Canada and the northern United States, and systematically overestimate amplitudes in other regions (e.g., Southeast USA, East Asia, and East Africa). We partially mitigated these discrepancies between model

outputs and observations by interpolating and smoothing the residuals, as commonly done for precipitation isotope maps to improve the fit of the maps to the data (e.g., Terzer et al., 2013). Better fits could have been achievable through using more predictor variables in the regression models, however we chose to limit the number of variables in the multiple regression models, even prior to the stepwise model selection; while we explored new relationships between precipitation isotope seasonality and (for example) diel temperature range or precipitation amount seasonality (Figure S1), these offer little

explanatory power that is not also captured in simpler metrics. Regardless, some uncertainties are introduced by using gridded climate products to infer site characteristics, because grid-cell means are not always representative of individual station

locations, as demonstrated by the mismatch between the elevations of monitoring stations and the mean elevations of the pixels they occupy (Figure S4). Other uncertainties in the regression predictions likely result from errors in the initial sine-curve fitting, as demonstrated by the fact that the regression models improve when only stations with longer records are used. For example, if we exclude all datasets shorter than three years (see Figure 2a), the $R^2$ of the $\delta^{18}O$ amplitude model increases from
0.64 to 0.73 and the $R^2$ of the offset model increases from 0.83 to 0.87. Any uncertainties in the models or the underlying data, however, do not preclude widespread estimation of precipitation stable isotope cycles at the level of confidence indicated (e.g., in Table 2 and Figure 5, or Figure 2b), which is improved upon through use of the residual-adjusted maps. Predictions can also be improved by using multiple regression models calibrated across individual regions of interest (using the data in Supporting Information 2).

These maps support predicting seasonal isotope cycles, but seasonal isotope cycles are only sometimes useful for predicting individual-month isotope values. To predict individual-month isotope values from a sine curve, the sine curve must be predictable (e.g., with well-constrained phase value), but also the sine curve must capture monthly isotope variations (e.g., $R^2$ must be high). In only a small subset of the monitoring stations were $R^2$ values consistently high (Figure 2c). For example, at
only 6 % of stations was more than 75 % of the variance explained by sine curves. Even fewer stations had long time series that enabled us to determine whether the high $R^2$ values also imply that inter-annual variations are small (e.g., such as in continental or northern latitude monitoring stations; Figure 2). Thus, individual month values should be carefully inferred from sine curves (e.g., by assuming errors of magnitudes like those shown in Figure 2b), even where precipitation isotope seasonality is predictable.

Precipitation isotope cycles are likely to be least predictable in latitudes near 30°S, 0°, or 30°N, where our models abruptly shift in phase, approximately demarcating global atmospheric circulation patterns. However, the inter-tropical convergence zone (ITCZ) is not consistently at 0° and Hadley cell boundaries are not consistently at 30°S and 30° N (in space or time; Birner et al., 2014; Chen et al., 2014), which may explain why most of the poor phase predictions (Figure 5b) occur near 30°
N or S. There are also errors near 0°, where predicted phase values differ by six months on either side of the equator, which does not precisely demarcate the ITCZ and relevant atmospheric circulations. Bowen et al. (2005) recognized this ITCZ effect and instead used the mean ITCZ position, rather than 0°, to account for phase shifts that occur there; although adopting Bowen's approach could mitigate some of the anomalies at 0° and 30° (Figure 5), other issues in predicting phase would persist (e.g., the elimination of higher frequency cycles; Jacobs et al., 2018). Thus, we opt for our simpler approach and accept that
our model is sometimes uncertain in zones near 0° and 30°, although those uncertainties are partially mitigated in the residual-adjusted maps.

Shortcomings in regression models may also result from not accounting for storm trajectories or convective effects, both of which influence precipitation isotope ratios (Aggarwal et al., 2016; Hu et al., 2018; Konecky et al., 2019). Models representing those processes can aid in interpreting or predicting stable isotope ratios (Hu et al., 2018; Risi et al., 2010). Furthermore, the variability in tropical precipitation isotopes ratios we show here may be the result of different storm sources and cloud types (Bailey et al., 2017; Scholl et al., 2009). Thus, precisely predicting precipitation isotope cycles in low latitudes without calibration data may (especially) require consideration of circulation patterns and their temporal variability (Cai et al., 2018; Martin et al., 2018b); an alternative option would be using regional multiple regression equations, which performed well in those regions (Figure 6). Regardless, most systematic effects should be compensated by the residual-smoothing step, as demonstrated by the relatively small prediction errors that we observed.

The 653 isotope monitoring stations used here span much of earth's climatic heterogeneity, but not all regions. The distributions of the site characteristics associated with these 653 monitoring stations are roughly similar to the global distributions of those characteristics (Figure S5). However, high-latitude, high-elevation monitoring stations are scarce (Figure S6). More notably, measurements are absent in large regions of Africa, Australia, central Asia, and north Asia. The most interior regions of continents generally contained the fewest monitoring stations (Figure 1b), and we suspect that our regression equations may underestimate the true increase in amplitude with distance from oceans (e.g., see amplitude underestimates in continental North America; Figure 4a). New precipitation isotope monitoring stations would help fill in important gaps.

These maps of seasonal precipitation isotope cycles serve as tools for studying terrestrial processes. In regions where seasonal precipitation isotope dynamics are well described by sine curves, sinusoidal isotope models are useful for predicting isotope values either at explicit points or continuously in time and space. The presence of large seasonal isotope cycles also enables the quantification of mixing, transport, and turnover of water (or its constituent O and H) in landscapes or biota. This is possible because 1) amplitude dampening reflects mixing processes, 2) phase shifts reflect advective travel times, and 3), offset differences reflect proportional contributions of different seasons' precipitation. In hydrology, the proportion of recent precipitation in streams can be estimated as the ratio of precipitation and streamwater isotope amplitudes (i.e., the young water fraction; Kirchner, 2016a). Maps of precipitation isotope cycles can facilitate estimating average precipitation amplitudes across catchments (Dutton et al., 2005; Jasechko et al., 2016). In such cases isotope values should ideally be weighted by precipitation amount, to diminish the influence of low volumes (von Freyberg et al., 2018). Quantifying seasonal precipitation isotope cycles also facilitates identifying the proportion (and over/under-representation) of precipitation from different seasons in samples such as surface waters (Bowen et al., 2019; DeWalle et al., 1997; Halder et al., 2015), groundwater (Jasechko et al., 2014; Kalin and Long, 1994; Lee and Kim, 2007), or plant and soil water (Allen et al., 2019). Similarly, ecological and physiological inferences can be drawn by observing how seasonal variations in water isotope signals are incorporated into (or propagate through) plant and animal tissues (Csank et al., 2016; Gessler et al., 2014; Martin et al., 2018a; Vander Zanden et

al., 2015; Yang et al., 2016). Even where phase values are poorly constrained, amplitude and offset values are still useful identifiers of typical mean values and magnitudes of seasonal variation. Thus, we expect that the mapped sine parameters that we have developed, as concise characterizations of seasonal precipitation isotope cycles, will find use in both physical and biological sciences.

These maps also indicate where precipitation isotope seasonality should be considered in interpreting isotopic signals in biological and geological samples. Annual mean precipitation may poorly predict the average isotopic input to any biological or geological process that does not integrate precipitation waters throughout entire years, particularly where precipitation isotopic composition is strongly seasonal (as discussed by, e.g., Dutton et al., 2005). Whereas event-to-event variations are likely to be rapidly damped by mixing in soils, lower-frequency variations, such as seasonal cycles, can persist and propagate through the water cycle. Where uptake and incorporation of isotopes into organisms (Balasse et al., 2003; Schubert and Jahren, 2015) or geologic materials (Johnson et al., 2006) also vary seasonally, mean annual precipitation may poorly and inconsistently approximate their average source water. For example, consider a hypothetical case of soil water with an isotopic composition that is consistently equal to that of the current month's mean precipitation. Further assume that a tree growing in this soil takes up that soil water and incorporates its oxygen atoms into cellulose during the six months of the warm season (e.g., when high-$\delta^{18}$O precipitation falls in high latitudes). For example, if the precipitation $\delta^{18}$O has a seasonal amplitude of 4 ‰, the average composition of the water taken up by the tree will be approximately $(2/\pi) \times 4$ ‰ $\approx 2.5$ ‰ higher than the annual average precipitation. This bias will be larger in locations where the seasonal amplitude of precipitation isotope cycles is larger. Thus, our maps showing precipitation isotope seasonality can be used to identify locations where such biases are potentially significant.

## 5 Summary

The majority of stable isotope time series measured at 653 precipitation isotope monitoring stations show significant sinusoidal seasonal cycles in precipitation isotopes. The fitted parameters that define these seasonal precipitation isotope cycles are estimated through multiple regression models of site characteristics. These spatial models enabled us to develop maps that describe global patterns in precipitation isotope seasonality, although regionally calibrated spatial models often better captured regional variations in precipitation isotope seasonality. The global maps and associated fitted isotope data are made available as supplementary information.

**Acknowledgements**

We thank the IAEA for developing and maintaining the Global Network for Isotopes in Precipitation (GNIP), and also thank the many researchers who have contributed data to GNIP. This project was funded by a grant from the Swiss Federal Office of the Environment to G.R. Goldsmith and J.W. Kirchner. Constructive comments were provided by three reviewers.

**Data Availability**

In Supporting Information 2, we provide all of fitted sine curves and site metadata for the 653 precipitation monitoring stations that are presented in this study. In Supporting Information 3, we provide metadata and a link to a 5-minute resolution gridded amplitude, phase, and offset for $\delta^{18}O$ and $\delta^2H$ of robust-fitted sine curves, hosted on Zenodo.org. All raw data used are synthesized from other studies or publicly available datasets; contact Dr. Jeff Welker regarding the USNIP (US Network for

Isotopes in Precipitation) dataset at: jmwelker@alaska.edu (the web site is under reconstruction).

**Supplementary Materials**

Supplementary Information 1

Table S1 Multiple regression coefficients and fit statistics for models describing the spatial variations in sine parameters that capture seasonal precipitation isotope cycles (amount-weighted fitted $\delta^{18}O$, robust-fitted $\delta^2H$, and amount-weighted fitted $\delta^2H$).

Fig. S1 Scatter plots of fitted sine parameters describing precipitation $\delta^{18}O$ seasonal cycles versus site characteristics that were not included in the regression models.

Fig. S2 Histogram of phase differences between seasonal isotope cycles and seasonal temperature cycles.

Fig. S3 Maps of fitted station values (markers) and the residual-adjusted maps of sine-curve parameters (shaded) that describe the seasonal cycles in precipitation $\delta^{18}O$ a) amplitude, b) phase, and c) offset.

Fig. S4 Elevations reported for sites regressed against gridded predictions of elevations of pixels containing those sites.

Fig. S5 Probability density functions of the site characteristics used to predict seasonal precipitation isotope cycles.

Fig. S6 Elevation versus latitude of precipitation isotope monitoring sites.

Supplementary Information 2

Data S1 Lists of fitted $\delta^{18}O$ and $\delta^2H$ sine parameter values (robust fitted, and amount-weighted fitted) and site characteristics

Supplementary Information 3

Data S2 Information to access the geospatial database (hosted on the public repository Zenodo.org), containing TIFF files of the following gridded products: $\delta^{18}O$ amplitude (‰); $\delta^{18}O$ phase (day of peak value in days from the summer solstice); $\delta^{18}O$

offset (‰); δ²H amplitude (‰); δ²H phase (day of peak value in days from the summer solstice); δ²H offset (‰); precipitation amount amplitude (mm/month); precipitation amount phase (day of peak value in days from the summer solstice); precipitation amount offset (mm/month).

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

**Table 1 Pearson and Spearman correlation coefficients of sine parameters versus (vs.) site characteristics.**

| Sine parameters | vs. \|latitude\| | vs. elevation | vs. dist. from coast | vs. temp. range | vs. mean temp | vs. mean precip. |
|---|---|---|---|---|---|---|
| **Pearson** | | | | | | |
| Amplitude | 0.34 | 0.34 | 0.54 | 0.58 | -0.56 | -0.35 |
| Phase | 0.76 | -0.12 | 0.25 | 0.72 | -0.68 | -0.64 |
| Offset | -0.67 | -0.16 | -0.23 | -0.70 | 0.88 | 0.40 |
| **Spearman** | | | | | | |
| Amplitude | 0.30 | 0.42 | 0.56 | 0.51 | -0.49 | -0.37 |
| Phase | 0.59 | 0.04 | 0.20 | 0.63 | -0.64 | -0.62 |
| Offset | -0.69 | -0.26 | -0.35 | -0.65 | 0.87 | 0.40 |

**Table 2 Multiple regression coefficients and fit statistics for models describing global variations in sine parameters that capture seasonal precipitation δ[18]O cycles. Dashes mark predictors that were excluded by the stepwise-regression model selection.**

| | \|Latitude\| (° from equator) | Elevation (m amsl) | Dist. from coast (km) | Temp. range (°C) | Mean Annual Temp. (°C) | Mean Annual Precip. (mm yr[-1]) | Intercept | RMSE | $R^2$ |
|---|---|---|---|---|---|---|---|---|---|
| Amplitude (‰ δ[18]O) | -0.06 | 0.0003 | 0.0013 | 0.08 | -0.12 | — | 4.5 | 1.1 | 0.64 |
| Phase (days)[a] | — | 0.005 | — | — | -0.38 | — | 24.2 | 12.0 | 0.19 |
| Phase (days)[b] | -1.27 | — | — | 0.78 | — | — | -100.0 | 28.2 | 0.21 |
| Offset (‰ δ[18]O) | 0.10 | — | — | -0.11 | 0.55 | -0.0008 | -15.7 | 2.0 | 0.83 |

[a] referring to sites in latitudes > 30° (N or S)

[b] referring to sites in latitudes < 30° (N or S)

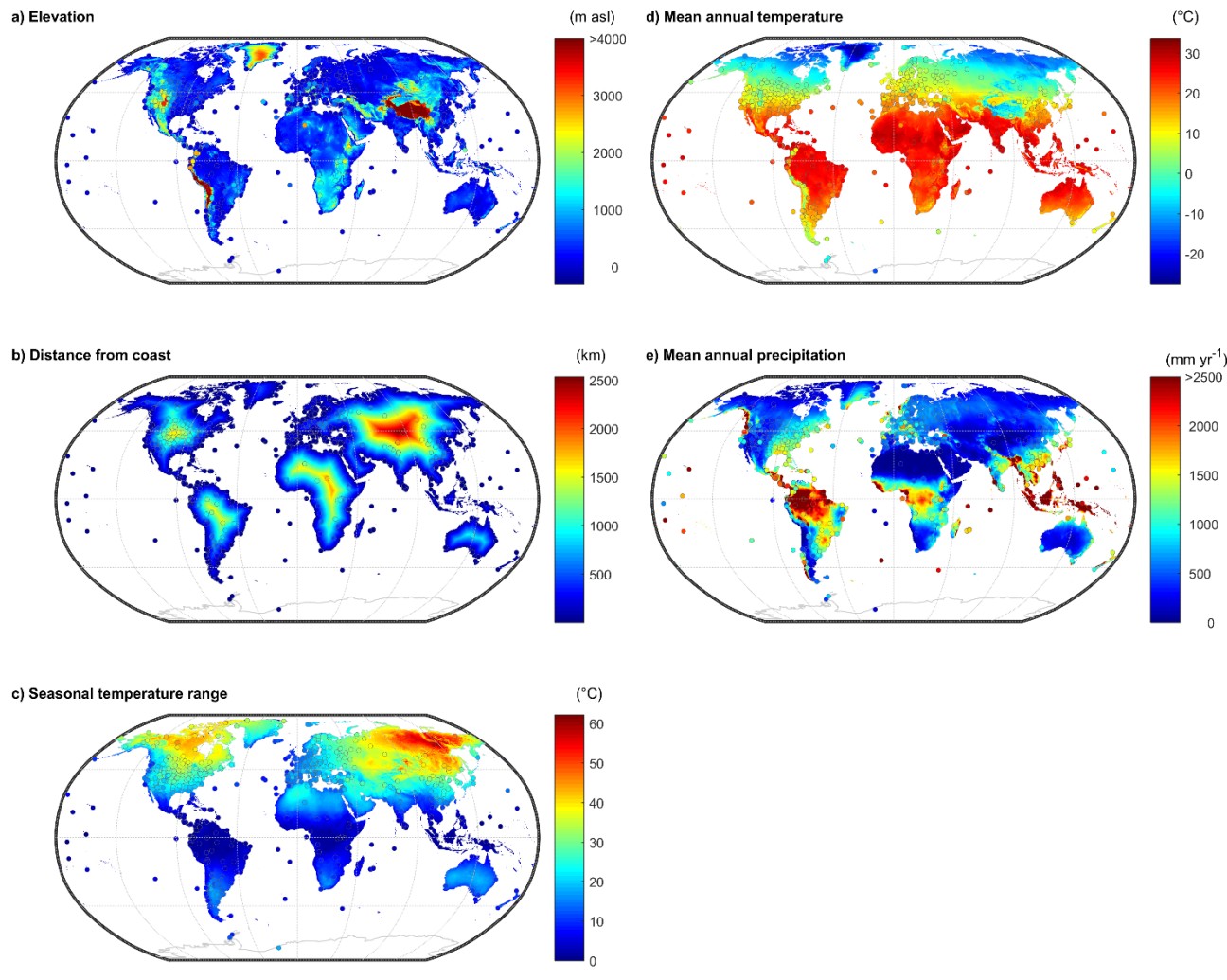

**Figure 1 Global maps of site characteristics used for predicting seasonal precipitation isotope cycles: a) elevation of precipitation isotope monitoring stations plotted over the elevation map, b) distance from coast, c) temperature range between mean temperatures of warmest and coldest months, d) mean annual temperature, and e) mean annual precipitation. Values at precipitation isotope monitoring stations are marked by circles. For b-e, station-level data are estimated as the value of the grid cells that the stations occupy.**

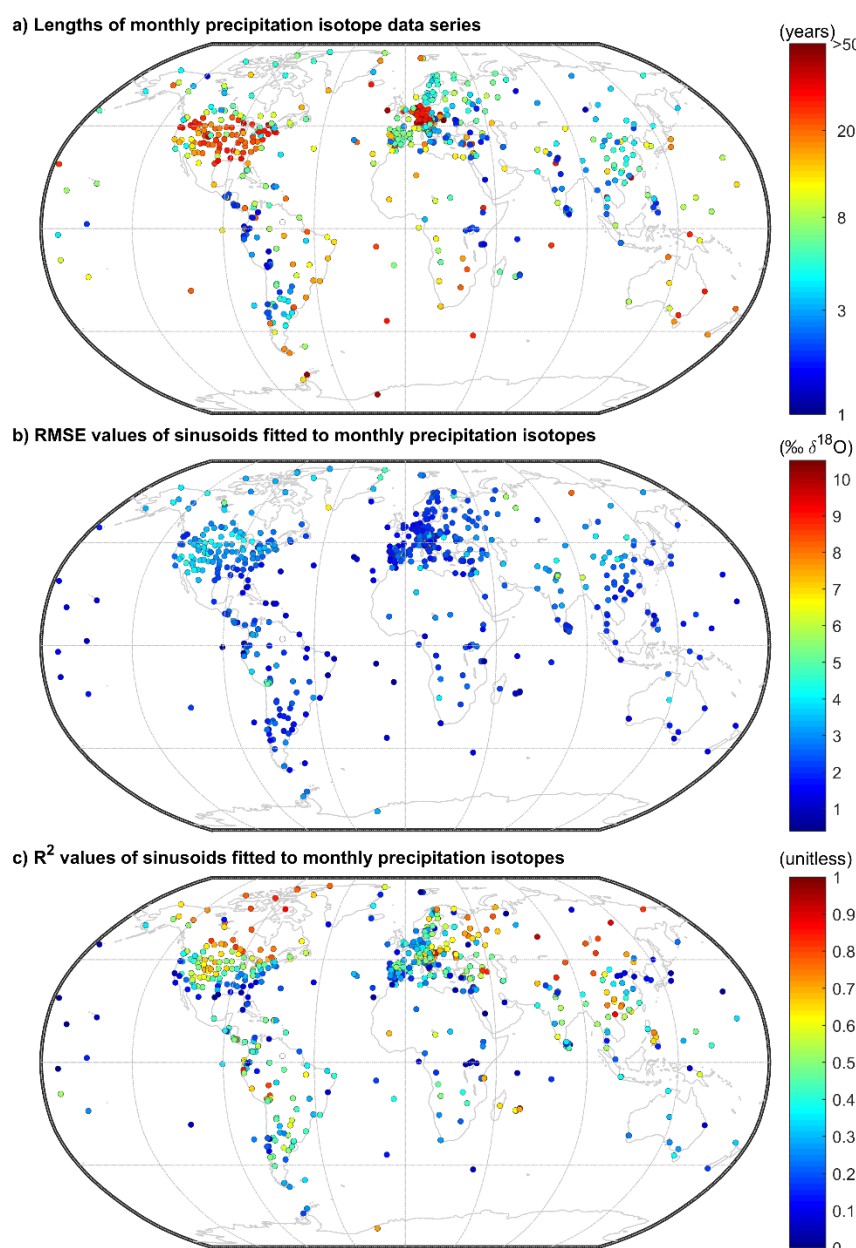

**Figure 2. Maps of precipitation isotope measurement stations with colours indicating a) the length of measurements at each site, and goodness-of-fit statistics b) root mean square errors (RMSE) and c) coefficients of variations ($R^2$) of the fitting of sine curves to monthly, empirical time series from each station. We show the robust-fitted $\delta^{18}O$ statistics; the amount-weighted $\delta^{18}O$ fit statistics, and the $\delta^2H$ statistics (robust-fitted and amount-weighted) are provided in the Supporting Information 2 data file.**

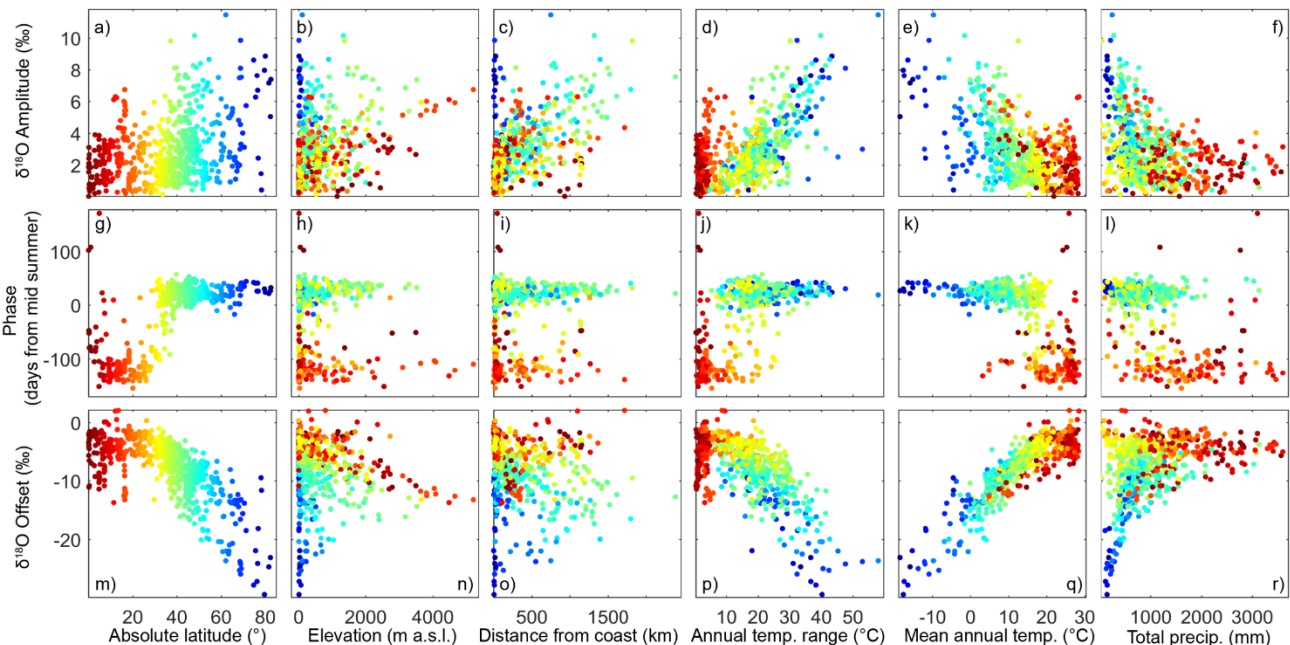

**Figure 3 Scatter plots of fitted sine parameters describing precipitation δ¹⁸O seasonal cycles – a-f) amplitude, g-l) phase, m-r) offset – versus site characteristics. For associated Spearman and Pearson correlation coefficients, see Table 1. Colours indicate absolute latitude (high latitudes in blue, low latitudes in red) as shown in panels a, g, and m.**

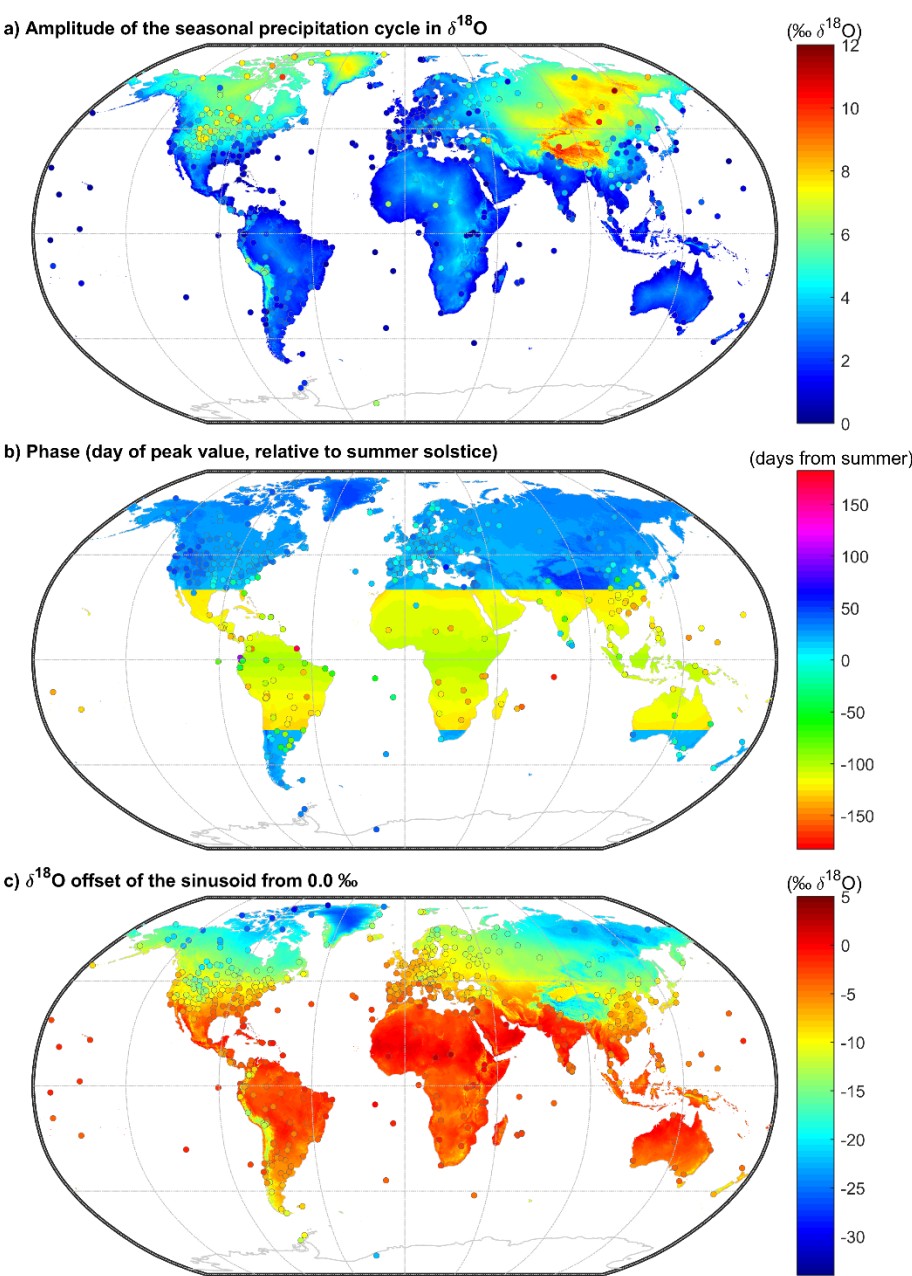

**Figure 4. Maps of fitted station values (markers) and regression-based sine-curve parameters (shaded) that describe the seasonal cycles in precipitation δ¹⁸O a) amplitude, b) phase, and c) offset. The shading reflects multiple-regression models based on landscape characteristics, described in Table 2; for phase, separate models were used in absolute latitudes > 30° versus latitudes < 30° (see methods). Here, residuals were not yet added back into the model.**

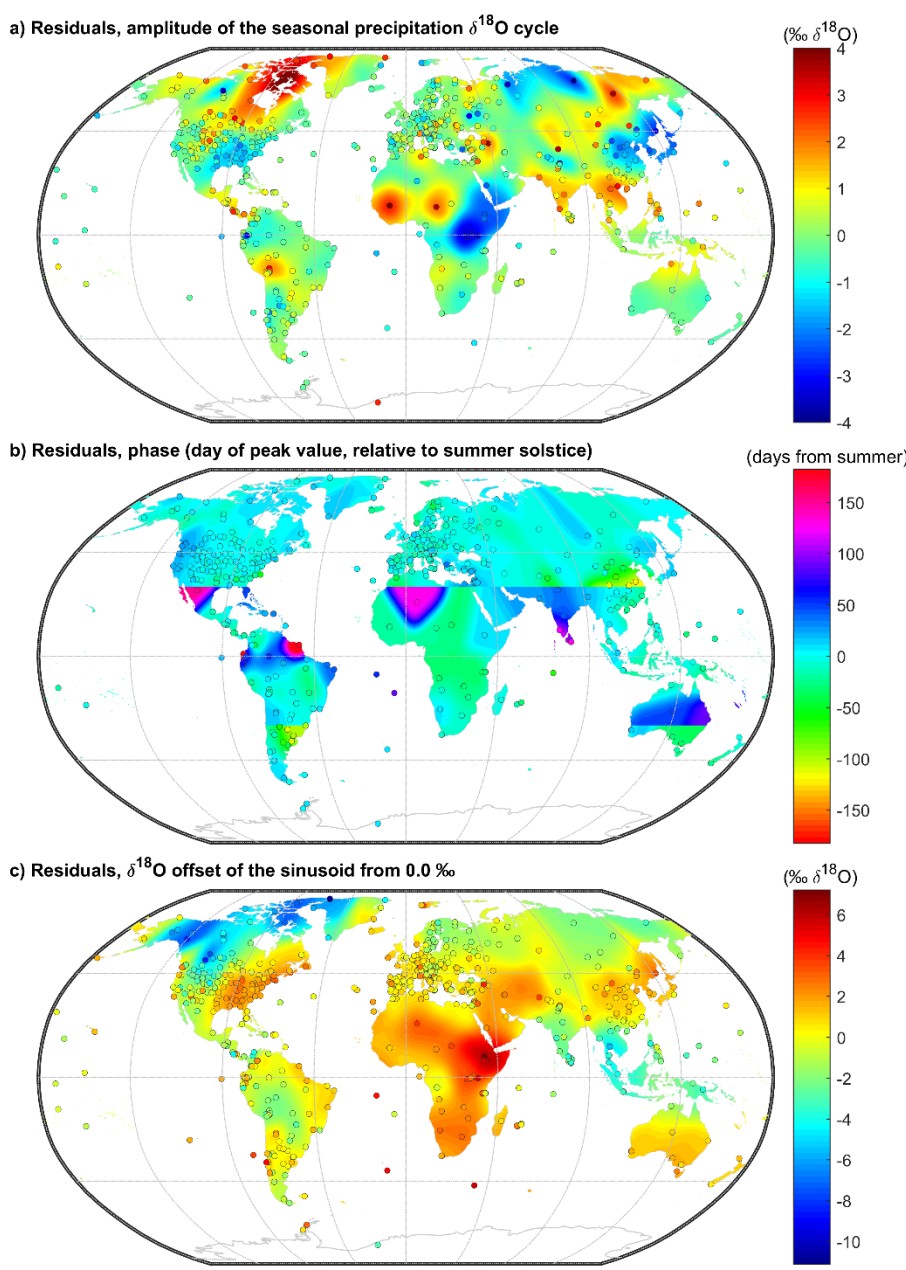

**Figure 5. Maps of δ¹⁸O a) amplitude, b) phase, and c) offset residuals, where the sine parameter values predicted from the multiple regression equations (shown in the interpolated maps in Figure 4) were subtracted from those of parameter values fitted to measurements at each precipitation isotope monitoring site (also shown in Figure 4). The shading shows the smoothed residual layers (see Methods).**

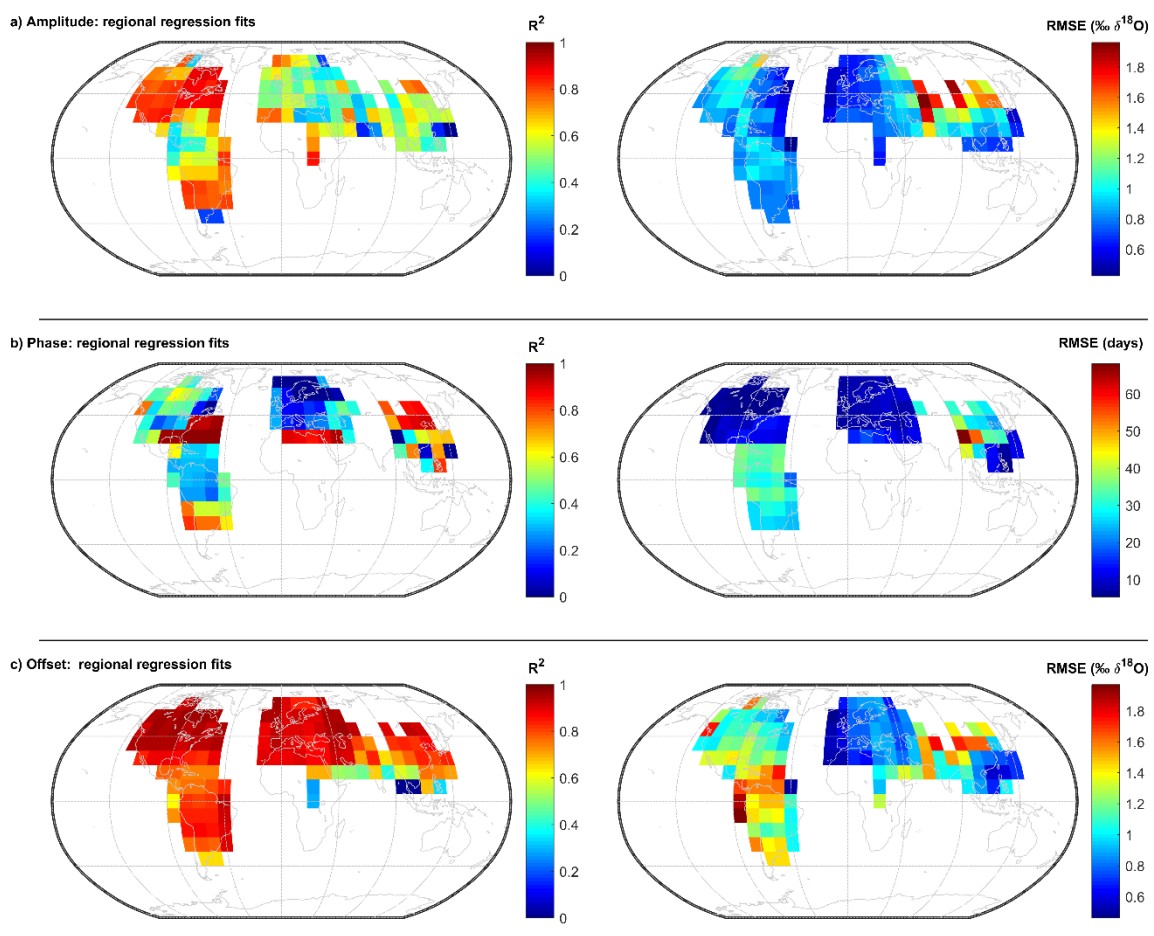

**Figure 6. Fit statistics for regionally fitted regressions that explain the spatial variations of the precipitation δ¹⁸O sine parameters. Regressions of a) amplitude, b) phase, and c) offset versus site characteristics were calculated for 40° × 40° pixels (centred on vertices at a 10° grid). Only pixels which contained >25 precipitation isotope measurements stations were used; for phase (b), we only used measurement stations that had well-constrained sinusoidal cycles (i.e., the standard error of the phase was less than 15 days). These figures show that site characteristics do not consistently explain the patterns of variations, and often the R² values are substantially lower than those of the global regression model (Table 2). However, the errors (RMSEs) are (almost) universally lower than those of the global regression model, implying that regionally calibrated regressions models are better predictors of spatial patterns in precipitation isotope cycles.**

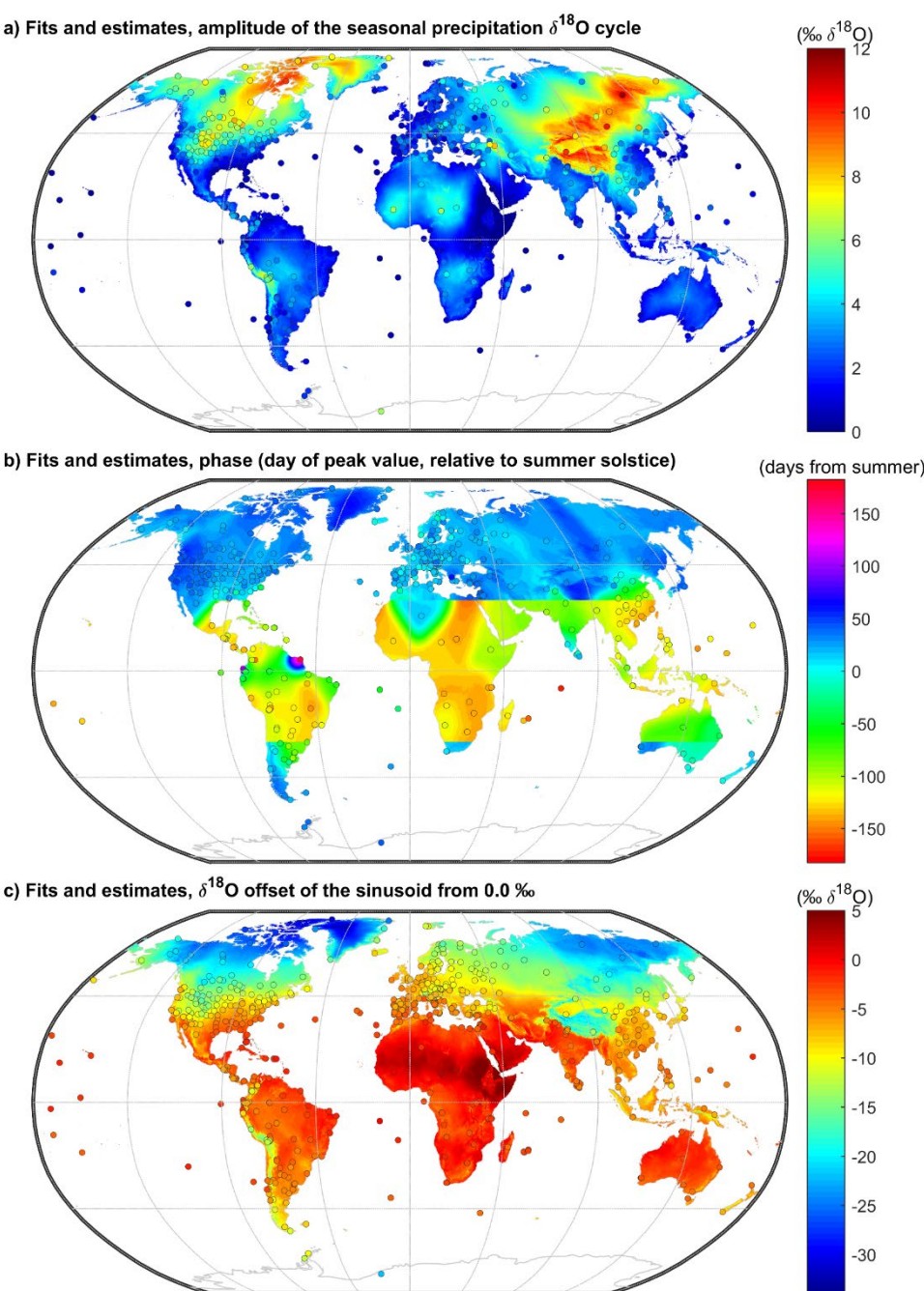

**Figure 7. Maps of fitted station values (markers) and the residual-adjusted maps of sine-curve parameters (shaded) that describe the seasonal cycles in precipitation $\delta^{18}$O: a) amplitude, b) phase, and c) offset. The interpolated surface is the sum of the infilled surfaces in Figures 3 and 4 (see Methods).**

