# Peer review of "Global sinusoidal seasonality in precipitation isotopes"

_Hydrology and Earth System Sciences, 2019_

## Referee Comment (RC1) · Anonymous Referee #1 · 5 Apr 2019

General comments: This manuscript describes a method to determine sine curve fits to the seasonal cycle of precipitation isotopes from stations around the globe. Interpolated maps of seasonality and a database of sine curve parameters were produced (not available for review). Overall the paper is well written, but ambitious in scope. The paper lacks an adequate explanation of how this work advances upon previous work, and needs more attention to sources of uncertainty in the analysis. With these improvements, the results presented here should be a solid contribution to the field of isotope hydrology.

Specific comments:

Abstract: this is somewhat disorganized, and would be improved by aiming toward a straightforward description of the problem or question addressed, the analyses done,

and the significance of the result.

P 2 L 4-10: Authors note that interpretive studies may ignore either the spatial or temporal aspect of the isotopic signal. Please explain further how the current approach improves on the interpolated seasonal data that are already available, where mean monthly isotope values can be downloaded from an online calculator for a set of spatial coordinates (isoscapes.org). The advancement represented by the approach in the current manuscript needs to be clearly described.

P 4 L 13-15; P10 L 13-14: Amount weighting is important for hydrological interpretations; please discuss whether amount is best included within the sine fitting procedure for an area, or whether amount should be included at the level of a regional or local study, where it would be used to weight the robust-fit seasonal values?

P 4 L 20: mean annual precipitation amount globally seems to have low predictive value for isotopic composition (table 2), does this parameter combine rainfall with snow water equivalent (SWE) measurements, and are those accurate enough to make this a useful parameter for station characterization?

P 5 L 1-2: are the areas and stations where there is no sinusoidal seasonal cycle clearly denoted in the database?

Section 2.3: Maps of predicted global precipitation isotope seasonality (sine curve parameters) and precipitation amount were generated with an interpolation scheme. Was any model validation performed by holding back a portion of station data and analyzing differences between measured and predicted isotopic value? This type of assessment should be done for the precipitation isotope seasonality and rainfall amount values.

P 6 L 15, L26-31, P7 L 1-5: It is not so surprising that tropical locations have seasonal cycles if one considers that land surface temperatures are not the primary control. Feng, X. et al., 2009, JGR, doi:10.1029/2008JD011279 (already cited); Scholl, M. et al., 2009, WRR, doi:10.1029/2008WR007515; Bailey, A. et al., 2017, JGR,

**HESSD**

doi:10.1002/2016JD026222 may provide a broader understanding of seasonal isotope patterns in the tropics. Condensation/equilibration temperatures can be very low and vapor sources isotopically depleted in tropical regions, where convective precipitation systems (esp. in the ITCZ) reach well above the freezing layer. The position of Hadley cell boundaries seems somewhat overemphasized here; atmospheric circulation factors that control isotope patterns (prevailing winds, atmospheric structure, dominant seasonal weather patterns) - have been identified in isotope-enabled GCM studies for tropical and temperate latitudes.

P 7 L 4: Precipitation d-excess globally exhibits a seasonal cycle, please see Pfahl and Sodemann, 2014, doi:10.5194/cp-10-771-2014. We would expect similar behavior for lc-excess, but with a less-distinct amplitude.

P 8 L 26-28: "grid-cell means are not always representative of individual station locations, as demonstrated by the mismatch between the elevations of monitoring stations and the mean elevations of the pixels they occupy (Figure S5)". Given that elevation is a major factor in isotopic composition of precipitation, how does this reflect on the interpolation and smoothing used to produce the maps? Should the map result be presented at the global scale, given that authors (appropriately) aim to "produce global maps and data that support stable isotope applications," and "predict individual-month values from a sine curve (P 9 L 6)"? Regional maps, where topography is presumably better represented, would seem to be a better approach and I encourage revision of this paper to include those maps and data sets, or at least a thorough explanation of the process of creating and calibrating regional maps.

P 10 L 5-7: Please identify "regions where "seasonal precipitation isotope dynamics are well described by sine curves," and where they are not, in a table or specific map. This would make the material much more informative to users of the data and prevent improper use of interpolated values. It is important to identify places where sinusoidal cycles cannot be used, especially given the discussion on p. 10 where authors suggest numerous applications for the data.

P 10 L 26: there are other references for this concept, please improve this section by including citations specific to the biological and geological processes that are noted; to improve the paper organization, consider moving material from lines 5-35 to the introduction, then briefly revisiting here.

P 11: "The *majority* of stable isotope time series measured at 653 precipitation isotope monitoring stations show significant sinusoidal seasonal cycles in precipitation isotopes" and "In Supporting Information 2, we provide fitted sine curves and site metadata for *all* 653 precipitation monitoring stations" . . . Given that some of the stations patterns do not have a sinusoidal cycle, why are sine curves being provided for stations where they are not applicable?

P 11 L 15-20: Supporting information 2 and 3 were not available for peer review and have not been evaluated. In this section, please provide details about the sources of raw data from "publicly available datasets" that were used in this work, with citations, attribution or links, to aid further research by others.

Figure S3 – this is not very informative at the coarse scale shown here - the reasons underlying phase shifts between temperature and isotopes (seasonality) globally are fairly well understood and should be addressed separately for different climate zones, if included at all. Figure 3b provides much the same information.
* * *

---

## Referee Comment (RC2) · Anonymous Referee #2 · 12 Apr 2019

General comments: This paper makes an important contribution to the scientific literature by providing estimates of coefficients of sinusoidal cycles in precipitation isotopic composition at global scale. These estimates are useful for analyses of water transit time and water source attribution in hydrological, biological, and geological studies. Regression models are presented that will allow users characterize precipitation isotope cycles at points or as raster grids.

Specific comments: P 2: additional information on previous geostatistical analyses (Bowen et al. 2014) and products, such as IsoMAP (http://isomap.org), should be included in the Introduction. Please explain how this study improves on previous work (eg., IsoMAP).

P 3 L 3-7: The data set used to develop the regressions is large and potentially very

useful to other users; however, a link to the data is not readily apparent. The authors indicate there is a compiled data set; however, I was unable to identify a link in the cited reference (Jasechko et al. 2016)(the Methods section of that paper indicates they compiled approximately 63K data points). Readers will not be able to reproduce the analysis in this paper without access to the precipitation isotope sample data. It is essential to provide a clear link to the raw data set (with appropriate citations for data sources).

P 4 L 19: the list of potential explanatory variables is reasonable; however, distance to nearest ocean ignores the influence of prevailing wind direction. While perhaps beyond the scope of this paper, it might be possible to include in future analyses. In the meantime, this source of error could be discussed in the Discussion section.

P 4 L 24-25: Model parameterization does not appear to follow accepted statistical best practices. In stepwise multiple regression, selection of model parameters usually is based on minimizing the Akaike information criterion (AIC) (Akaike 1981) or Bayesian information criterion (BIC), rather than maximizing $R2$, which could lead to model over-parameterization. Colinearity does not appear to have been considered quanitatively, but should be; it often is tested using the variance inflation factor (VIF) (Hair et al. 2005).

P 7 L 4: define LC-excess.

P 8 L 10-15: One of the main contributions of this paper is the presentation of models for amplitude, phase, and offset. This allows readers to estimate these cycle characteristics at other sites and/or create raster grids (as the authors have done). This is worth mentioning explicitly in the Discussion.

P 9 L 3-4: cannot locate Supporting Information 2.

Table 1: consider including p-values.

References: Akaike, Hirotugu. 1981. "Likelihood of a model and information criteria."

Journal of Econometrics 16 (1):3-14. doi: 10.1016/0304-4076(81)90071-3.

Bowen, Gabriel J, Zhongfang Liu, Hannah B Vander Zanden, Lan Zhao, and George Takahashi. 2014. "Geographic assignment with stable isotopes in IsoMAP." Methods in Ecology and Evolution 5 (3):201-206.

Hair, J.F., W.C. Black, B. Babin, R. Anderson, and R.L. Tatham. 2005. Multivariate data analysis. 6th ed. Upper Saddle River, New Jersey: Prentice Hall.

Jasechko, Scott, James W. Kirchner, Jeffrey M. Welker, and Jeffrey J. McDonnell. 2016. "Substantial proportion of global streamflow less than three months old." Nature Geosci 9:126-129. doi: 10.1038/ngeo2636.
* * *

---

## Referee Comment (RC3) · Anonymous Referee #3 · 23 Apr 2019

The authors present an incredibly useful predictive statistical model of the global patterns of d18O and d2H in precipitation. The methods are adequate and sound and the results are clearly described and presented in tables and figures. In my opinion, the manuscript can be accepted in its present form. I leave the following three comments only to encourage the authors to expand the discussion if they agree it would improve the paper.

The authors' objective to produce the predictive model was clearly motivated by a need in the hydrological community for isotopic input data to calculate young water fractions and unravel storage selection behavior of watersheds using stable isotope data. The observed patterns in explanatory variables are only lightly discussed in terms of atmospheric circulation patterns or origins of atmospheric water vapor.

There are a number of studies that have used atmospheric air mass trajectory analyses to study the variability of isotopes in precipitation. I understand this is well outside the scope of this manuscript - and possibly out of reach computationally. It might be worth mentioning air mass trajectory analysis as a possible path for improving the predictions of stable isotopes in precipitation.

On page 4, the authors describe the decision to use the "robust-fitted" seasonal parameters (as opposed to the "amount-weighted" parameters) for further analysis because they capture the variations during drier seasons better. I wonder if the "amount-weighted" offset would provide a better estimate which is less biased by light (summer) precipitation events and if there is a significant difference between the two estimates.

---

## Author Comment (AC1) · 29 May 2019

The authors present an incredibly useful predictive statistical model of the global patterns of d18O and d2H in precipitation. The methods are adequate and sound and the results are clearly described and presented in tables and figures. In my opinion, the manuscript can be accepted in its present form. I leave the following three comments only to encourage the authors to expand the discussion if they agree it would improve the paper.

The authors' objective to produce the predictive model was clearly motivated by a need in the hydrological community for isotopic input data to calculate young water fractions and unravel storage selection behavior of watersheds using stable isotope data. The observed patterns in explanatory variables are only lightly discussed in terms of atmospheric circulation patterns or origins of atmospheric water vapor.

There are a number of studies that have used atmospheric air mass trajectory analyses to study the variability of isotopes in precipitation. I understand this is well outside the scope of this manuscript - and possibly out of reach computationally. It might be worth mentioning air mass trajectory analysis as a possible path for improving the predictions of stable isotopes in precipitation.

The reviewer is correct that our primary objective is motivated by needs of the hydrologic community. The reviewer is also correct that air-mass trajectory effects could result in some of the scatter in the initial regression models. We now add further discussion on air-mass trajectory effects.

On page 4, the authors describe the decision to use the "robust-fitted" seasonal parameters (as opposed to the "amount-weighted" parameters) for further analysis because they capture the variations during drier seasons better. I wonder if the "amountweighted" offset would provide a better estimate which is less biased by light (summer) precipitation events and if there is a significant difference between the two estimates.

Although we do focus on the robust fitted data in the manuscript, we also provide values for the amount-weighted fits as part of the data products provided. They can be directly compared using the supplemental data that we will now provide. We also have extended the methods section, where we describe the two fitting approaches, to emphasize that these two metrics have different limitations.

---

## Author Comment (AC2) · 30 May 2019

Response to Referee #2

We thank the reviewer for their effort put towards reviewing our manuscript. Our responses below are in black, Time's New Roman and referee's comments are in blue, Calibri font.

General comments: This paper makes an important contribution to the scientific literature by providing estimates of coefficients of sinusoidal cycles in precipitation isotopic composition at global scale. These estimates are useful for analyses of water transit time and water source attribution in hydrological, biological, and geological studies. Regression models are presented that will allow users characterize precipitation isotope cycles at points or as raster grids.

Specific comments:

P 2: additional information on previous geostatistical analyses (Bowen et al. 2014) and products, such as IsoMAP (http://isomap.org), should be included in the Introduction. Please explain how this study improves on previous work (eg., IsoMAP).

We now further discuss other isotope data products. It is important to note that our analysis yields a very different product: maps that show seasonal cycles, rather than predictions of isotope values in specific months or years (e.g., products from Bowen et al.). Statistically, our approach first extracts the seasonal signal from the data, and then interpolates those signals. As such, the values used in the interpolations are a product of an entire time series, not just single points.

Our paper allows readers to immediately understand the location and strength of seasonal cycles in precipitation isotopes. Perhaps most importantly, our method provides information (i.e. sine curve parameters) that is not directly available from isomap.org or the online isotopes in precipitation calculator (OIPC), but is increasingly used in isotope hydrology (e.g., young water fraction calculations). While products from Bowen et al. could be used for alternative calculations of isotope seasonality, that product is not currently available. We are not critiquing Bowen et al.'s method, we are simply offering a different product for use in hydrological analyses and expect that the product will find uses beyond its obvious intended applications.

We now make these points in the introduction.

P 3 L 3-7: The data set used to develop the regressions is large and potentially veryuseful to other users; however, a link to the data is not readily apparent. The authors indicate there is a compiled data set; however, I was unable to identify a link in the cited reference (Jasechko et al. 2016)(the Methods section of that paper indicates they compiled approximately 63K data points). Readers will not be able to reproduce the analysis in this paper without access to the precipitation isotope sample data. It is essential to provide a clear link to the raw data set (with appropriate citations for data sources).

We now cite the data sources in the data table and update the data availability statement so that it specifies how all of the data can be accessed. It is true that these precipitation data were previously analyzed by Jasechko et al. and were obtained via direct download from the IAEA's database (ref. 34 in Jasechko et al. 2016 and http://www-naweb.iaea.org/napc/ih/IHS_resources_isohis.html) and via personal communication with leaders of two national precipitation isotope networks: S. J. Birks (e.g. see ref. 37 in Jasechko et al. 2016) and J. M. Welker (e.g. see ref. 36 in Jasechko et al. 2016). By providing the fitted sinusoid statistics, this paper marks a step forward because it does provide a single compiled dataset of metrics describing precipitation isotope data.

P 4 L 19: the list of potential explanatory variables is reasonable; however, distance to nearest ocean ignores the influence of prevailing wind direction. While perhaps beyond the scope of this paper, it might be possible to include in future analyses. In the meantime, this source of error could be discussed in the Discussion section.

We now expand our discussion of how circulation patterns and storm trajectories relate to isotope patterns.

P 4 L 24-25: Model parameterization does not appear to follow accepted statistical best practices. In stepwise multiple regression, selection of model parameters usually is based on minimizing the Akaike information criterion (AIC) (Akaike 1981) or Bayesian information criterion (BIC), rather than maximizing R2, which could lead to model overparameterization. Colinearity does not appear to have been considered quantitatively, but should be; it often is tested using the variance inflation factor (VIF) (Hair et al. 2005).

We understand that AIC is commonly used, but in our case, we found that minimizing the AIC led to the selection process retaining more terms than were retained by our method; we now mention this in the manuscript. Note that our method was not solely to maximize $R^2$ values, because we also excluded all coefficient p-values that were not statistically significant ($p < 0.05$). We now report the VIFs. Even if we hypothetically used all of the potential predictor values (which was never the case), all of VIFs are less than 10 (i.e., a commonly used cutoff value).

P 7 L 4: define LC-excess.

LC-excess is now defined (and Landwehr and Coplen 2006 is cited).

P 8 L 10-15: One of the main contributions of this paper is the presentation of models for amplitude, phase, and offset. This allows readers to estimate these cycle characteristics at other sites and/or create raster grids (as the authors have done). This is worth mentioning explicitly in the Discussion.

We now more clearly emphasize this point.

P 9 L 3-4: cannot locate Supporting Information 2.

We opted to not include these until the final version is accepted because of the possibility that reviewers might ask us to alter our algorithm. This supplement will now be included.

Table 1: consider including p-values.

Given the size of the dataset, we prefer to not include p values because they are all extremely small ($p=10^{-6}$ for the weakest of these regression).

References:
Akaike, Hirotugu. 1981. "Likelihood of a model and information criteria."
Journal of Econometrics 16 (1):3-14. doi: 10.1016/0304-4076(81)90071-3.

Bowen, Gabriel J, Zhongfang Liu, Hannah B Vander Zanden, Lan Zhao, and George Takahashi. 2014. "Geographic assignment with stable isotopes in IsoMAP." Methods in Ecology and Evolution 5 (3):201-206.

Hair, J.F., W.C. Black, B. Babin, R. Anderson, and R.L. Tatham. 2005. Multivariate data

analysis. 6th ed. Upper Saddle River, New Jersey: Prentice Hall.

Jasechko, Scott, James W. Kirchner, Jeffrey M. Welker, and Jeffrey J. McDonnell. 2016. "Substantial proportion of global streamflow less than three months old." Nature Geosci 9:126-129. doi: 10.1038/ngeo2636.

---

## Author Comment (AC3) · 30 May 2019

General comments: This manuscript describes a method to determine sine curve fits to the seasonal cycle of precipitation isotopes from stations around the globe. Interpolated maps of seasonality and a database of sine curve parameters were produced (not available for review). Overall the paper is well written, but ambitious in scope. The paper lacks an adequate explanation of how this work advances upon previous work, and needs more attention to sources of uncertainty in the analysis. With these improvements, the results presented here should be a solid contribution to the field of isotope hydrology.

Thank you for your feedback. We will now better explain the differences between our research and others' related research. To clarify, our paper allows readers to immediately understand the location and strength of seasonal cycles in precipitation isotopes. Perhaps most importantly, our method provides information (i.e. sine curve parameters) that is not directly available from isomap.org or the online isotopes in precipitation calculator (OIPC), but is increasingly used in isotope hydrology (e.g., young water fraction calculations).

In the revised draft, we better explain what this data product offers and what limitations it has (see comments below as well as responses to reviewer 2). Indeed sine curves do not perfectly represent precipitation isotope variability; they are, however, a useful metric to visualize and describe presence, strength, and timing of seasonality. Understanding where these patterns occur aids in guiding future studies' analytical approaches. We have also added an additional uncertainty analysis (see comments below).

Specific comments:

Abstract: this is somewhat disorganized, and would be improved by aiming toward a straightforward description of the problem or question addressed, the analyses done, and the significance of the result.

In rereading our abstract, we realize that some confusion could arise because it is hard to discern between results and methods (because the paper does focus on methodology). We believe that this is warranted because we are producing a data product, and thus the results often justify subsequent methodological steps. Nonetheless, we have revised the manuscript to make it adhere to a more typical structure.

P 2 L 4-10: Authors note that interpretive studies may ignore either the spatial or temporal aspect of the isotopic signal. Please explain further how the current approach improves on the interpolated seasonal data that are already available, where mean monthly isotope values can be downloaded from an online calculator for a set of spatial coordinates (isoscapes.org). The advancement represented by the approach in the current manuscript needs to be clearly described.

In our method, we first capture the seasonal cycle, and then interpolate. We now specify that difference and its importance as an advance on current methods. Secondly, our analysis is a next-level data product, mostly intended for specific hydrology applications (although others are discussed later). One can quickly look up how amplitudes vary using our study, which would require many steps

using isomap.org data. Since publishing Allen et al 2018, many people have asked for the product that we are here providing because completing these sine fits can be tricky (e.g., with respect to constraining phase values, consistent amplitude definitions, and quantifying errors). We provide these geospatial data with the hope and expectation that other researchers can complete their own studies more efficiently.  We have better clarified our specific objectives in the abstract, introduction, and conclusions.

 P 4 L 13-15; P10 L 13-14: Amount weighting is important for hydrological interpretations; please discuss whether amount is best included within the sine fitting procedure for an area, or whether amount should be included at the level of a regional or local study, where it would be used to weight the robust-fit seasonal values?

The amount weighting can be important, but we prefer not to say which approach is "best" because the two support different applications. One scenario where amount-weighting is important in the fitting is if there was an anomalously dry month (e.g., 1 small precipitation event), where the storm had atypically high δ18O values, or the sample was exposed to evaporation in the collector. In this scenario, the amplitude would be exaggerated in a non-weighted sine fit. If these values were later weighted a typical amount, it could result in a misrepresentation of the precipitation inputs. In an alternative hypothetical scenario, using weighted fits in a Mediterranean region (with dry summers) might under-represent the true seasonal amplitude. Both metrics are valuable, and we will now further discuss their respective limitations in the methods section (2.2).

P 4 L 20: mean annual precipitation amount globally seems to have low predictive value for isotopic composition (table 2), does this parameter combine rainfall with snow water equivalent (SWE) measurements, and are those accurate enough to make this a useful parameter for station characterization?

Precipitation amount varies by orders of magnitude across the globe; we are only using 1$^{st}$-order linear regressions, so it is not overly surprising that it does not explain much (and is thereby mostly excluded from the regression equations). As discussed in response to reviewer 2, there is indeed some collinearity among these predictor variables.

While quantifying snowfall inputs can be challenging, it is unlikely to cause errors that are large compared to the range of variation in precipitation amounts across sites. We now specified that SWEs are accounted for.

P 5 L 1-2: are the areas and stations where there is no sinusoidal seasonal cycle clearly denoted in the database?

No they were not, but they will now be made identifiable because the phase term will be replaced with "NA"; the amplitudes and offsets are still useful (as will be described in the manuscript).

Section 2.3: Maps of predicted global precipitation isotope seasonality (sine curve parameters) and precipitation amount were generated with an interpolation scheme. Was any model validation performed by holding back a portion of station data and analyzing differences between measured and predicted isotopic value? This type of assessment should be done for the precipitation isotope seasonality and rainfall amount values.

We remind the reviewer that we are presenting this as a method for predicting seasonal cycles, not for predicting individual monthly values (unlike the sinusoidal model used in Allen et al 2018, which was tested using individual months). We now validate the model's prediction of sine parameters. Individual month values are highly erratic and can deviate substantially from the sine curves (as we show in what is now Figure 2, and formerly S1); there is a whole paragraph dedicated to discussing this point. We have now re-run the model iteratively, holding back subsets of the sites to be used

solely in validation and not in calibration. We will report those values as prediction errors of amplitude, phase, and offset.

P 6 L 15, L26-31, P7 L 1-5: It is not so surprising that tropical locations have seasonal cycles if one considers that land surface temperatures are not the primary control. Feng, X. et al., 2009, JGR, doi:10.1029/2008JD011279 (already cited); Scholl, M. et al., 2009, WRR, doi:10.1029/2008WR007515; Bailey, A. et al., 2017, JGR,doi:10.1002/2016JD026222 may provide a broader understanding of seasonal isotope patterns in the tropics. Condensation/equilibration temperatures can be very low and vapor sources isotopically depleted in tropical regions, where convective precipitation systems (esp. in the ITCZ) reach well above the freezing layer. The position of Hadley cell boundaries seems somewhat overemphasized here; atmospheric circulation factors that control isotope patterns (prevailing winds, atmospheric structure, dominant seasonal weather patterns) - have been identified in isotope-enabled GCM studies for tropical and temperate latitudes.

Thank you for directing us to these papers, which will now be cited in our study. We will expand our discussion of previously described tropical isotope cycles and the role circulation patterns in driving those cycles.

P7L4: Precipitation d-excess globally exhibits a seasonal cycle, please see Pfahl and Sodemann, 2014, doi:10.5194/cp-10-771-2014. We would expect similar behavior for lc-excess, but with a less-distinct amplitude.

Yes, it is well described that seasonal d-excess cycles exist (and we will now cite this reference). However, while d-excess variations can result from variations along a LMWL of slope < 8, LC-excess variations result from systematic seasonal deviations from LMWLs. Thus, we are describing patterns that differ from what is described previously in Pfahl and Sodemann. We will now expand on this difference to clarify.

P 8 L 26-28: "grid-cell means are not always representative of individual station locations, as demonstrated by the mismatch between the elevations of monitoring stations and the mean elevations of the pixels they occupy (Figure S5)". Given that elevation is a major factor in isotopic composition of precipitation, how does this reflect on the interpolation and smoothing used to produce the maps? Should the map result be presented at the global scale, given that authors (appropriately) aim to "produce global maps and data that support stable isotope applications," and "predict individual-month values from a sine curve (P 9 L 6)"? Regional maps, where topography is presumably better represented, would seem to be a better approach and I encourage revision of this paper to include those maps and data sets, or at least a thorough explanation of the process of creating and calibrating regional maps.

The answer to this question obviously depends on application. We shared these same concerns, which is exactly why we pursued this project as we did: a) using a high resolution DEM to conduct the initial interpolation, b) producing the global map, but also carefully showing where it fails to capture individual points, c) processing and sharing the data such that it can be immediately incorporated into regional regression models and d) showing where regional models perform best (Figure 6). Of course, we cannot produce every regional map because we cannot anticipate every future 'region' for which a map might be needed.

Regarding the point-to-cell mismatch, this problem is true of most exercises in which data collected at one scale are interpreted as representative of a larger scale. By conducting the new error analysis described above, we quantify the magnitude of error which is partially due to the problem described here.

We also further emphasize to readers that the ability to "predict individual-month values" from sine curves requires that the $R^2$ values are high, and not just that there is a seasonal cycle.

P 10 L 5-7: Please identify "regions where "seasonal precipitation isotope dynamics are well described by sine curves," and where they are not, in a table or specific map. This would make the material much more informative to users of the data and prevent improper use of interpolated values. It is important to identify places where sinusoidal cycles cannot be used, especially given the discussion on p. 10 where authors suggest numerous applications for the data.

Previously, information showing how well sine curves capture the monthly variations is in Figure S1. We now move Figure S1 into the main text so that nobody misses this information (Now Figure 2). We now also further describe the limitations and their consequences on Page 10.

P 10 L 26: there are other references for this concept, please improve this section by including citations specific to the biological and geological processes that are noted; to improve the paper organization, consider moving material from lines 5-35 to the introduction, then briefly revisiting here.

We do not agree that it is helpful to lengthen the introduction to then just revisit those points later, as this information is not critical for understanding the basis of the study. Nonetheless, we have improved on this by including relevant citations in the discussion.

P 11: "The *majority* of stable isotope time series measured at 653 precipitation isotope monitoring stations show significant sinusoidal seasonal cycles in precipitation isotopes" and "In Supporting Information 2, we provide fitted sine curves and site metadata for *all* 653 precipitation monitoring stations" ... Given that some of the stations patterns do not have a sinusoidal cycle, why are sine curves being provided for stations where they are not applicable?

This is an excellent point and we will now comment on this in the second-to-last paragraph in the discussion section. Even where there is not a sinusoidal cycle, the sine curve provides a measure of central tendency, which is of value. Also, by providing the sine-fit and parameters, we can show why the curve fit is not 'significant': e.g., short time series, small amplitude, erratic month-to-month values (i.e., leading to a large RMSE). Furthermore, including near-zero amplitudes can be important for fitting regional regression models. The phase term is the only one of the three parameters that is meaningless when the sine curve is not significant. We remove the phase values of non-significant sine curves from the data tables because we cannot think of any application for which those would be useful.

P 11 L 15-20: Supporting information 2 and 3 were not available for peer review and have not been evaluated. In this section, please provide details about the sources of raw data from "publicly available datasets" that were used in this work, with citations, attribution or links, to aid further research by others.

We now provide references in the supporting information table.

Figure S3 – this is not very informative at the coarse scale shown here - the reasons underlying phase shifts between temperature and isotopes (seasonality) globally are fairly well understood and should be addressed separately for different climate zones, if included at all. Figure 3b provides much the same information.

This is in the supplemental because we also believe that it is not crucial, but potentially helpful or interesting to someone. Thus, we prefer to keep it, but we can enhance it by adding two more panels to show the effect of latitude on these differences.